# Marker-Assisted Improvement for Durable Bacterial Blight Resistance in Aromatic Rice Cultivar HUR 917 Popular in Eastern Parts of India

**DOI:** 10.3390/plants12061363

**Published:** 2023-03-18

**Authors:** Manish Kumar, Ravi Pratap Singh, Debarchana Jena, Vineeta Singh, Diptibala Rout, Panduranga Bhagwan Arsode, Madhu Choudhary, Prakash Singh, Suman Chahar, Sanghamitra Samantaray, Arup Kumar Mukherjee, Chander Mohan, Abhishek Bohra, Goutam Das, Sumana Balo, Onkar Nath Singh, Ramlakhan Verma

**Affiliations:** 1Institute of Agricultural Science, Banaras Hindu University, Varanasi 221005, Uttar Pradesh, India; 2ICAR-National Rice Research Institute, Cuttack 753006, Odisha, India; 3Veer Kunwar Singh College of Agriculture, Bihar Agricultural University-BAU, Sabaur, Dumraon, Buxar 802136, Bihar, India; 4Central State Farm, National Seeds Corporation, Sirsa Road, Hisar 125001, Haryana, India; 5Department of Agriculture, Cooperation and Farmers Welfare, Government of India, New Delhi 110001, Delhi, India; 6State Agricultural Biotechnology Centre (SABC) and Centre for Crop and Food Innovation (CCFI), Murdoch University, Perth, WA 6150, Australia; 7Department of Soil Science and Agricultural Chemistry, Uttar Banga Krishi Vishwavidyalaya, Coochbehar 736165, West Bengal, India; 8Birsa Agricultural University (BAU), Ranchi 834006, Jharkhand, India

**Keywords:** marker-assisted selection, biotic stress, BB resistance gene, broad spectrum durable resistance, ASG rice, yield and quality

## Abstract

Bacterial blight (BB) is a devastating disease of rice in the tropics of Indian sub-continent, where the presence of *Xoo* races with varying levels of genetic diversity and virulence renders disease management extremely challenging. In this context, marker-assisted improvement of plant resistance has been proven as one of the most promising approaches for the development of sustainable rice cultivars. The present study demonstrates the marker-assisted introgression of the three BB resistant genes (*Xa21 + xa13 + xa5*) into the background of HUR 917, a popular aromatic short grain (ASG) rice cultivar in India. The performance of the resulting improved products (near isogenic lines (NILs), HR 23-5-37-83-5, HR 23-5-37-121-10, HR 23-5-37-121-14, HR 23-65-6-191-13, HR 23-65-6-237-2, HR 23-65-6-258-10 and HR 23-65-6-258-21) establishes the utility of marker-assisted selection (MAS) approach for accelerated trait introgression in rice. The MAS-bred lines carrying three introgressed genes showed broad spectrum BB resistance (lesion length, LL of 1.06 ± 1.35 cm to 4.61 ± 0.87 cm). Besides, these improved lines showed the complete product profile of recurrent parent HUR 917 along with the enhanced level of durable BB resistance. The improved introgression lines with durable BB resistance would contribute to sustainable rice production in India, particularly in the Indo-Gangetic plane that has substantial acreage under HUR 917.

## 1. Introduction

Rice has the global identity as staple food crop for more than half of the world population with substantial acreage (167.13 mha) and production (782.0 million tons) [1]. In India, it is grown on ~44.5 mha area producing 172.5 million tons with an average productivity of 3.88 tons per hectare [1] which is directly linked to the livelihood and economy. Global rice production should increase by 40% to feed the world population by 2030 [2]. The sustainability of the global rice production is challenged by several factors, of which growing vulnerability of the crop to changing pest pathogen dynamics remains crucial [2,3]. More than 40 types of biotic and abiotic stresses threaten rice cultivation around the globe. The key biotic stresses of rice include diseases caused by various fungi, bacteria, nematode and viruses that impact upon almost every stage of the crop growth and development [3]. Amongst these diseases, bacterial blight (BB) (*Xanthomonas oryzae* pv. *Oryzae)*, blast caused (*Magnaporthe oryzae*) and sheath blight (ShB) (*Rhizoctonia solani*) substantially reduce rice yield in addition to deteriorating the quality of the rice grain [3]. 

Indian sub-continent is well known for its native wealth of basmati and aromatic non-basmati rice, of which aromatic short grain (ASG) types remain important with respect to aroma, cooking and quality traits. These traits are responsible for greater consumer preference globally and fetching premium prices in domestic as well as in global markets [4,5]. Unfortunately, low yielding capacity and susceptibility to various pests and diseases render the deployment of ASG types in rice breeding challenging. Non-basmati scented rice varieties are primarily grown in the Indo-Gangetic region of India including states like Uttar Pradesh, Bihar, and Chhattisgarh. The HUR 917, a popular ASG type rice variety of medium duration, has raised much hope amongst farmers and rice-exporters owing to its superiority over other preferred aromatic non-basmati rice [5,6]. However, the susceptibility of this rice variety to BB contributes to inconsistent yield.

BB is a devastating disease across rice ecology (temperate to tropical) and territory, causing substantial losses (20–100%) in rice yield [7,8]. Disease symptoms appear in the form of leaf blight and *Kresek* (wilt phage). The warm temperatures (25–30 °C), high humidity, heavy rainfall and water stagnation coupled with heavy nitrogen dose [9] promote disease occurrence. Besides, luxuriant crop growth, close spacing and winds injury also make rice crop prone to BB spread [10]. The *Xoo* inoculum enters through injured plant epidermis and perpetuates in the xylem vessels. The pathogenic *Xoo* races, highly dynamic and diverse in nature [11,12,13], produce a variety of toxic secretions including extracellular polysaccharides, extracellular enzyme, iron chelating siderophores and the type III-secretion dependent effectors [14,15] that compromise plant’s immune response [16]. To date, globally, 30 races of *Xoo* have been documented [13,17,18].

Improving plant resistance through incorporation of resistance genes (R genes) is an effective strategy for managing BB in rice [19]. Modern genomic tools and techniques have facilitated mapping and more importantly, the stacking of R genes into elite agronomic backgrounds having susceptibility the disease [11,20,21,22,23]. More than 40 genes have been reported in rice that confer BB resistance with varying range of effectiveness against diversified *Xoo* races [22,24]. Of these, the genes, *Xa4*, *xa5*, *Xa7*, *xa13, Xa21* and *Xa38* have been most frequently deployed to incorporate BB resistance in rice cultivars [20,21,22]. For instance, the *xa5* gene encoding gamma subunit of transcription factor IIA5 (TFIIA) in combination with the genes *Xa7* and *Xa21* improved BB resistance of the recipient genotypes [20,25,26]. Similarly, the gene *xa13* encoding a plasma membrane protein has been reported to confer broad spectrum BB resistance, in complementation with *Xa21* against majority of the Indian virulent *Xoo* races [20,27]. The *Xa21* gene coding for an NBS-LRR protein remains the most effective for imparting broad spectrum BB resistance and therefore the *Xa21* gene has been extensively used in rice breeding programs [20,21,22,28,29].

The functional complementation of these gene(s) is found to be more effective, and results in comparatively broader spectrum of durable resistance in comparison to the deployment of the single resistance gene [20,21,22,30]. The eastern region of India is the hotspot for BB disease, and the R genes with functional complementarities including *Xa21 + xa13 + xa5 + Xa4*, *Xa21 + xa13 + xa5* and *Xa21 + xa13* have been proven successful against the *Xoo* races [20,21,22]. Therefore, deployment of functional R gene combinations is imperative to extend the durability and resistance levels to counter the evolving virulent races of the pathogen [31,32]. Recent advancements in molecular breeding techniques make it convenient to improve the varieties/lines for desirable traits with great precision and efficiency. Molecular breeding techniques such as MAS, marker-assisted backcrossing (MABC), and genomic selection (GS) have provided strong evidence in support of their efficient use in indirect selection/or to trace the trait of interest in plant breeding programs [20,33]. 

In view of the above, the current study was undertaken with the objective to transfer R gene combinations with the greatest functional compatibility (*Xa21 + xa13 + xa5*) into a BB-susceptible yet popular short grain aromatic rice variety HUR 917. The study incorporated broad spectrum BB resistance into the rice variety HUR 917 that is still grown in sizeable acreage in Indo-Gangetic planes of India (Figure 1).

## 2. Materials and Methods 

### 2.1. Experimental Materials and Breeding Strategy

We selected IRBB66 carrying five BB resistance genes (*Xa21 + xa13 + Xa7 + xa5 + Xa4*) and having closest genetic relationship (0.77) with the recurrent parent (RP) for marker-aided improvement of BB resistance level of the popular variety HUR 917 (Table 1 and Appendix A) [34]. Hybridity of the individual F_1_ plants generated from the cross HUR 917/IRBB66 was confirmed with the DNA markers (pTA248, xa13prom and RM122) [3,7,20,35,36,37] (Table 2). The true hybrids were backcrossed with the RP to generate BC_1_F_1_ seeds. Advancement of target gene(s)/traits with maximum RP genome recovery was assessed in every back-cross (BC) generation with linked and informative DNA markers, followed by phenotyping. The best BC_1_F_1_ plants with maximum RP genome and phenome were advanced to the next generation. The BC_2_F_1_ plants were also subjected to foreground selection (FS), background selection (BS) and phenotyping to identify the plants with maximum recovery for RP genome and phenome (Table 3 and Appendix A). Selection differential (Δd) analysis for the product profile traits, days to fifty percent flowering (DFF), plant height (PH), grain L/B ratio, head rice recovery (HRR) and aroma content was performed in BC and segregating generation (Appendix A). The percent disease index (PDI) and area under disease progress curve (AUDPC) analysis was done to recover BB resistant NILs (Appendix A). The positive BC_2_F_1_ with perfect product profile traits were advanced to BC_2_F_2_ generation, and FS was repeated in order to identify the plants that were homozygous for *Xa21 + xa13 + xa5* gene combination (Table 3). Further, desirable plants were advanced with single seed descent (SSD) under field using rapid generation advance (RGA) strategy to accelerate development of the NILs [36].

### 2.2. PCR and Marker Analysis

The genomic DNA from the leaf sample of each plant (20–22 days old seedling) was extracted and purified using CTAB method [39]. The PCR reaction was carried out using 15 ng of template DNA, 1× Taq assay buffer, 0.3 mM of MgCl_2_, 133.0 µM of dNTPs, 1 U/µL of Taq DNA polymerase (Thermo-scientific, Life Science Products, Mumbai, Maharashtra, India) and 1.25 µM of each primer (Eurofins, Genomics, Hyderabad, India). The PCR was carried out in an Eppendorf Thermo Cycler with the following program: (1) initial denaturation at 94 °C for 3 min; (2) 39 cycles for denaturation for 30 s at 94 °C, annealing for 30 s at 56 °C, extension for 1 min at 72 °C; and (3) final extension at 72 °C for 5 min. The amplified products were resolved with 2.5% Metaphor^TM^ Agarose gel (Typhoon FLA 700, Alpha Innotech, MA, USA) and visualized under a UV light source in photographed gel documentation (Gel-Doc) system (Gel Doc^TM^ XR, Bio-Rad Laboratories Inc., Hercules, CA, USA).

Target traits in F_1_ and BC generations were traced using trait linked-DNA markers, pTA248 lying 0.2 cM distant to *Xa21* [7,36,37], xa13prom with 0 cM distant to *xa13* [3,20,36] and RM122 with 0.4 cM distance to *xa5* [35,36] (Table 2). Moreover, genome recovery of the RP was monitored with 82 polymorphic SSR markers (Appendix A) that span entire rice genome [20,33,40]. These 82 SSRs were selected from 360 SSRs based on the polymorphism survey between HUR 917 and IRBB66. Recombinant selection was carried out using the markers RM26969 (*Xa21*), RM23356 and RM22914 (*xa13*) and RM17941 (*xa5*) flanking the BB resistant genes on chromosome 11, 8, and 5, respectively, (Table 2). The molecular data in the backcross generations were used to visualize genotype of individuals using Graphical Genotyper (GGT2.0) software [41]. 

### 2.3. Disease Bioassay Analysis

Each BC generations and NILs (BC_2_F_3_) carrying effective hetero/homo-alleliec combinations of R genes were grown in field condition along with parents and bio-assayed with eight virulent *Xoo* pathotypes/races (*Xa17, Xa7, xa2, Xb7, Xc4, xd1, xa1* and *xa5*) prevalent in the eastern region of the country. The pathotypes were maintained in peptone sucrose agar (PSA) medium [42] and single spore culture with 10^8^ cfu/mL bacterial *Xoo* suspension density was used for inoculation. The top five leaves of each plant were clipped off and inoculated [43]. Post inoculation, the plants were observed after every 24 hours’ time interval to note the appearance of disease symptoms. The LL were measured at 14, 21 and 28 days after inoculation (DAI) [13] using a disease score index of 0–9 [44,45]. The LL of <5 cm was considered resistant (R), 5–10 cm was considered moderately resistant (MR), 10–15 cm was moderately susceptible and >15 cm was considered highly susceptible. The epidemiological parameters like PDI and AUDPC which indicates overtime disease accumulation were calculated to assess the disease severity [9]. 

### 2.4. Agro-Morpho Evaluation of NILs

The BC_2_F_3_ and BC_2_F_4_ NILs along with RP and donor parents were evaluated for yield and other agro-morphological traits in randomized complete block design (RCBD) with three replications. Standard agronomic management practices were followed for raising the rice crop at two locations (ICAR-National Rice Research Institute, Cuttack and Banaras Hindu University, Uttar Pradesh) in India. The data were recorded on five plants from each of the entries for the characters namely: days to panicle initiation (DPI), days to first panicle emergence (DFPE), days to 50% flowering (DFF), days to maturity (DM), plant height (PH), number of effective tillers per plant (NETPP), panicle length (PL), number of grains per panicle (NGP), test weight (TW), grain yield per plant (GYPP), and disease severity (DS). Further, the lines were also analyzed for grain and cooking quality parameters such as head rice recovery (HRR) [46], kernel length before cooking (KLBC), kernel breadth before cooking (KBBC), length/breadth ratio (L/B), amylose content (AC), GC content [47] and aroma as described by [48]. The statistical analysis was performed using DARwin-6.0, XLSTAT and SPSS packages. Use of plant material complies with relevant institutional, national, and international guidelines and legislation.

## 3. Results

### 3.1. Molecular Characterization of Parents and Selection of Donor 

Of the total 360 SSRs screened among the six rice genotypes (HUR 917, IRBB66, IR 64 MAS, CRMS 31B MAS, improved PR 114 and Lalat MAS), 82 (22.8%) SSRs showed polymorphism among the RP, HUR 917 and the five donors (Appendix A). The diversity analysis revealed varying degrees of genetic relatedness (0.06 to 0.77) among the parents (Appendix A). The donor genotype, IRBB66 (*Xa21 + xa13 + Xa7 + xa5 + Xa4*) showed closest genetic relation with RP (0.77) followed by IR 64-MAS (*Xa21 + xa13 + xa5 + Xa4*) (0.76) and CRMS 31B-MAS (*Xa21 + xa13 + xa5 + Xa4*) (0.50). The bioassay analysis of IRBB 66 also showed incompatible or HR disease reaction (PDI of 3.65 ± 0.110, AUDPC of 74.13 and LL of 0.57 ± 0.21 cm to 1.83 ± 0.35 cm) (Table 4 and Appendix A). To avoid undesirable linkage drag and allow a rapid fixation of segregating loci, the genotype IRBB66 [22] was used as a donor in the present backcross scheme. 

### 3.2. MABB Based Trait Improvement for BB Resistant in HUR-917

The F1s of HUR 917/IRBB66 were tested for hybridity using trait-linked SSR markers pTA248 (~0.2 cM, from *Xa21*) [7,36,37], xa13prom (0.0 cM from *xa13*) [3,20,36] and RM122 (0.4 cM from *xa5*) [35,36] (Table 2). The five F_1_s with confirmed hybridity were then crossed with the RP to generate BC_1_F_1_s that were evaluated for agro-morphological traits. Eighteen of the total 240 BC_1_F_1_ plants were found to carry three R alleles (*Xa21 + xa13 + xa5*) with 69.51 to 81.71% RP genome and superior phenome (Agro-morphology* and moderate resistant (MR), PDI of 11.82 ± 0.360 to 35.14 ± 0.344, AUDPC of 204.43 to 599.27) (Table 3 and Appendix A, Figure 2) [49]. Six BC_1_F_1_s carrying >80.0% RP genome and desirable phenome were advanced to BC_2_F_1_ with 265 seeds (data not presented). Three positive BC_2_F_1_ plants carrying 90.85 to 91.46% RP genome and superior phenome (at par with RP with PDI of 5.19 ± 0.114 to 13.48 ± 0.752, MR) were self-pollinated to develop NILs (Table 3, Appendix A, Figure 3 and Figure 4) [49]. The BS revealed heteroallelism for 26.9 and 9.24% SSR markers in BC_1_F_1_ and in BC_2_F_1_, respectively, which were further analyzed in BC_2_F_2_ (Appendix A). FS analysis in 740 BC_2_F_2_ seeds revealed homozygosity for R genes in 15 plants (2 and 3 R genes), with disease reactions confirming their resistant to moderately resistant nature (PDI of 2.48 ± 0.624 to 18.24 ± 0.225, AUDPC of 72.24 to 178.25) (Table 4 and Appendix A). The BS analysis in BC_2_F_2_ with SSR markers suggested >92.0% RP genome recovery in BC_2_F_2_ (Table 3 and Table 4). BC_2_F_2_ plants were subjected to phenotyping and generation advancement to allow selection of the most desirables.

### 3.3. Genome Introgression on the Carrier and Non-Carrier Chromosomes

The success of the trait improvement strategy relies upon the extent of the RP genome recovery in the resulting products while retaining originality of the RP. Our results suggested an average of 95.17% RP genome recovery in the resulting pyramided lines, representing genomic segments from the chromosomes 1, 2, 4, 5, 6, 7, 9, 10 and 12 (Table 3 and Figure 5). The R genes (*Xa21*, *xa13* and *xa5*) located on chromosomes 11, 8 and 5, respectively, were staked with negligible linkage drags (delimited up to 3.4-Mb downstream of *xa5*, 2.6-Mb upstream and 4.1-Mb downstream of *xa13*, and 1.1-Mb upstream of *Xa21*) (Figure 5, Appendix A). Analysis with the SSR marker RM122 supported successful introgression of *xa5* in total 14 NILs along with >80% RP genome recovery. Similarly, the marker pattern of *xa13* prom suggested the presence of *xa13* in eleven NILs with 85.7% RP genome, where HR 23-5-37-83, HR 23-5-37-109, HR 23-5-37-121, HR 23-65-6-38, HR 23-65-6-191, HR 23-65-6-237, HR 23-65-6-242, HR 23-65-6-298 and HR 23-135-83-24 were recovered with the complete chromosome 8 of the RP parent. Besides, 11 NILs having *Xa21* had substantial RP genome (83.2%) with desirable phenotypic traits. On an average, 76.07 (72.75%) markers confirmed homozygosity and 3.00 (3.65%) markers had heterozygosity, whereas 2.93 (3.57%) markers showed homozygosity for donor alleles. The pyramids (BC_2_F_2_), HR 23-135-83-24, HR 23-65-6-258 and HR 23-5-37-201 with all target R genes were recovered with 95.12% RP genome followed by HR 23-5-37-83, HR 23-65-6-142 and HR 23-65-6-237 with 93.90% RP genome and phenome (Appendix A and Figure 5).

### 3.4. Morphological Evaluation of NILs

The phenotyping data of selected plants in backcross generations were recorded progressing but least selection differential (Δd) for most of the product profile traits like, days to 50% flowering (DFF) (2.12), plant height (PH) (1.68), grain L/B ratio (0.15) and head rice recovery (HRR) (1.53) (Appendix A). Analysis of variance (ANOVA) among NILs and parents showed substantial variances (at *p* ≤ 0.001) in all studied traits (Table 5). The selected BC_2_F_4_ NILs, HR 23-5-37-121-10 (7132.62 kg/ha), HR 23-65-6-237-2 (6692.66 kg/ha), HR 23-5-37-83-5 (6399.36 kg/ha), HR 23-65-6-258-10 (6066.06 kg/ha), HR 23-65-6-258-21 (5866.08 kg/ha), HR 23-65-6-191-13 (5499.45 kg/ha) and HR 23-5-37-121-14 (5399.46 kg/ha) carrying targeted R gene combination (*Xa21 + xa13 + xa5*) showed broad spectrum of BB resistance (PDI of 3.28 ± 0.24 to 7.42 ± 0.35, LL of 1.06 ± 1.35 cm to 4.61 ± 0.87 cm) (Table 4, Table 6 and Appendix A, Figure 4, Figure 5, Appendix A). These selected BC_2_F_4_ NILs were subjected to robust phenotyping for grain quality, and the results revealed similar quality as of RP with selection differentials: Δd = 0.11 (Grain L/B ration), Δd = 0.72 (HRR) and Δd = 0.0 (Aroma) (Table 6, Appendix A). The majority of the three resistance genes containing NILs were similar to the RP for the basic agro-morpho and quality traits (Figure 6). Furthermore, genetic relatedness amongst parents and NILs were assessed based on similarity metrics data of 82 SSR markers, which ranged from 0.06 to 0.94 (data not presented). The entire 17 genotypes (including parents) could be distinguished into 2 two major clusters, cluster I-A consisted of only one genotype i.e., IRBB66 with similarity coefficient value of 0.77. Whereas cluster-II consisted of RP and 15 NILs derivatives with maximum similarity coefficient value of 0.94 between HUR 917 (RP) and HR 23-5-37-201 followed by HR 23-5-37-109 (0.93), HR 23-135-83-95 (0.93), HR 23-5-37-83 (0.91) and HR 23-5-37-121 (0.93). The NILs, HR 23-5-37-83-5, HR 23-5-37-121-10, HR 23-5-37-121-14, HR 23-65-6-191-13, HR 23-65-6-237-2, HR 23-65-6-258-10 and HR 23-65-6-258-21 which are carrying triplet homoalleles of R genes (*Xa21 + xa13 + xa5*) and had more than 90% RP genome achieved complete product profile as of HUR 917 (Table 6 and Appendix A). The HR 23-5-37-121-10, HR 23-65-6-258-21 and HR 23-5-37-121-14 were found to have similar 1000-grain weight (Test weight), HRR, AC content and aroma as of RP which are the most important product profile trait needed to be recovered in this experiment (Table 6, Appendix A). The quality assessment of NILs suggested that the introgressed R gene combination had similar quality parameters including grain type (Table 6 and Appendix A) HRR (53.6% to 79.2%), AC content (20.09% to 24.12%), and GC content (31 to 59) (Table 5). It is important to note that the aroma (BADH2), which was fixed in early generation, was retained in the NILs (Appendix A). Besides, sensory evaluation results revealed that there was no significant difference between RP and NILs with respect to aroma. (Appendix A).

### 3.5. Bioassay of the NILs for BB Resistance 

The results from bioassay of NILs suggested resistance reactions (PDI of 1.83 ± 0.24 to 9.12 ± 0.37, AUDPC of 36.07 to 154.5, LL of 1.06 ± 1.35 cm to 4.61 ± 0.87 cm), and the recorded levels remained at par with the donor parent (Table 4 and Appendix A). Fifteen NILs namely, HR 23-5-37-83-5, HR 23-5-37-83-12, HR 23-5-37-121-3, HR 23-5-37-121-10, HR 23-5-37-121-14, HR 23-5-37-201-9, HR 23-65-6-142-3, HR 23-65-6-142-18, HR 23-65-6-191-13, HR 23-65-6-237-2, HR 23-65-6-237-27, HR 23-65-6-258-10, HR 23-65-6-258-21, HR 23-135-83-24-1 and HR 23-135-83-24-22 that attained homozygosity for the three R genes had broad spectrum of BB resistance against eight *Xoo* pathotypes (Table 4, Figure 4). The NILs (BC_2_F_3_s) harboring three homo-alleles of R genes (*xa21 + xa13 + xa5*) and maximum RP genome (93.90 to 94.51%) had LL in the range of 1.06 ± 1.35 cm to 4.61 ± 0.87 cm, which corresponded to resistant category (disease score of 1). The HR 23-5-37-83-5 showed LL between 1.37 cm to 2.72 cm and product profile at par with the RP for the traits, DFF (108 days), TW (16.4g), HRR (68.6%), AC (22.73%) and aroma (2), followed by HR 23-5-37-121-10 (LL, 1.34 cm to 2.17 cm; DFF, 110 days; TW, 14.2 g; HRR, 65.8%; AC, 22.87% and aroma, 2), HR 23-5-37-121-14 (LL, 1.97 cm to 4.03 cm; DFF, 108 days; TW, 13.6g; HRR, 76.5%; AC, 24.12% and aroma, 2), HR 23-65-6-191-13 (LL, 2.27 cm to 3.19 cm; DFF, 110 days; TW, 16.4g; HRR, 53.6%; AC, 23.66% and aroma, 2 score), HR 23-65-6-237-2 (LL, 1.70 cm to 2.62 cm; DFF, 105 days; TW, 16.4g; HRR, 53.6%; AC, 23.66% and aroma, 2 score), HR 23-65-6-258-10 (LL, 1.03 cm to 4.61 cm; DFF, 109 days; TW, 16.4 g; HRR, 73.1%; AC, 22.78% and aroma, 2 score) and HR 23-65-6-258-21 (LL, 1.06 cm to 3.03 cm; DFF: 106 days; TW, 15.6 g; HRR, 79.2%; AC, 20.12% and aroma, 2 score) (Table 4, Table 6 and Appendix A, Figure 4). The LL observed on the lines containing *Xa21 + xa5* genes varied from 4.20 ± 1.23 cm to 7.97 ± 1.16 cm, while these ranged from 1.74 ± 0.51 cm to 2.53 ± 0.31 cm in the NILs carrying *xa13 + xa5* gene combination (*data are not presented). 

## 4. Discussion

Non-basmati ASG rice with exclusivity to aroma, has excellent cooking quality traits, owing to which it attracts global consumer preference and premium prices in the international markets [4,5]. It is also a reservoir of nutritionally important traits. Research aiming to improve nutrients like Zn, Fe, vitamins and antioxidants by utilizing this genetic resource has opened the new avenues for rice export [50]. Despite having comparatively low amylose content, ASG lines provide better quality and aroma than basmati types [51]. In addition, this genetic resource retains substantial aroma even under warmer conditions [6]. Widespread preference for the ASG variety HUR 917 faces some challenges due to its less yielding capacity [52], vulnerability against stresses [53] and lodging [54]. The increased BB susceptibility of this variety has caused inconsistency in its yield. Enhancing resistance level of the host plant presents a standard practice to mitigate the risks of disease and pest attacks [20,21,22,54]. Evidence suggests that stacking of more than one R gene resulting in quantitative complementation or synergistic response, improves the resistance level of plants [55]. In recent years, advanced genomic tools [5,33,54,56] have made significant contributions to trait improvement in plant breeding. Several R genes conferring durable resistance against major biotic stresses have been successfully incorporated in basmati and non-basmati rice backgrounds [18,20,21,22,48,57].

In this study, the genotype IRBB 66 (a derivative of IR 24) has high level of resistance against hyper-virulent *Xoo* pathotypes owing to the presence of five R genes (Table 4 and Figure 4). We selected IRBB 66 as a donor for MABC scheme because of its highest genetic proximity to the RP (Figure 2) and similarity in quality traits of both [20]. The F1s with hetero-alleles of effective R gene combinations (*Xa21+ Xa7 + Xa4*) had resistant reaction, however, backcross progenies with only *Xa21* gene had moderately resistant reaction. The improved lines did not show any linkage drag from the donor genomic segments (Figure 5 and Figure 6), which might be due to genetic closeness among donor and RP genotypes. DMRT and biplot analysis revealed perfect recovery of product profile and value-added traits in the derivatives (Table 6 and Figure 6) [36]. Stepwise BS analysis coupled with robust phenotyping led to a substantial increase in the RP genome (94.51% in BC_2_F_2_ and 96.95% in BC_2_F_4_^*^) [21,22]. The majority of the NILs carrying three R genes (*Xa21 + xa13 + xa5*), were found to be morphologically similar (duration, grain, yield and quality parameters) to RP with least selection differential (>0.90 similarity index, data not presented) (Table 3, Table 6 and Appendix A) and at par resistance with the donor, IRBB66 (Table 4). Phenotyping in combination with MAS in backcross generations was proven equally important as this strategy helped fast-track the transfer of genomic regions of functional relevance. Whereas SSR markers employed in the BS usually target the non-coding and heterochromatic regions and therefore may not be suitable to quantify the recovery of functional part of the genome [21]. (*data are not presented).

Notably, selection differentials for most of the desirable traits changed progressively throughout BC and segregating generations. No change in aroma, a key parameter, was reported in the BC_1_F_1_ generation (Score-02) (Appendix A). Intensive selection caused fixing of the aroma trait (a key trait in ASG governed by the two recessive genes *badh1* and *badh2*) during the early generations [58,59,60,61]. The sensory panel test revealed no substantial differences for aroma in RP and the derivatives, which validated the successful recovery of all major gene(s) responsible for fragrance i.e., 2-acetyl-1-pyrroline (2AP) in rice. The targeted product profile in NILs was attained since most of the traits considered in the study were attributed to a number of small effect genes/QTLs. 

Adoption of limited backcrosses in this study found to have substantial edge over burden of more backcrosses as our rapid backcross scheme accelerated accumulation of desirable genes (minor) in derivatives. Thus, inclusion of genetically close parents in MAS not only reduces the breeding cycle, but it can also contribute to enhance response to selection for important traits in later stage of the population [21,62,63].

The bioassay of NILs confirmed the successful introgression of the R genes, and the lines manifested incompatible disease response against all virulent *Xoo* pathotypes considered in the study. The improved introgression lines with two or more effective R gene combinations showed resistance levels comparable to that of donor, which validated the efficacy of the introgression strategy followed in the current study. 

## 5. Conclusions

Genomics-aided trait improvement to combat BB disease is an environmentally-sound and economically-viable strategy to achieve sustainable rice production worldwide. Prevalence of genetically diverse virulent *Xoo* strains over agro-climatic zone renders this disease extremely challenging to manage. The results of this study revealed that three R genes (*xa5* + *xa13* + *Xa21*) in combination could impart broad spectrum BB resistance. We demonstrated the precise stacking of R genes into the background of the popular rice variety HUR 917. 

## Figures and Tables

**Figure 1 plants-12-01363-f001:**
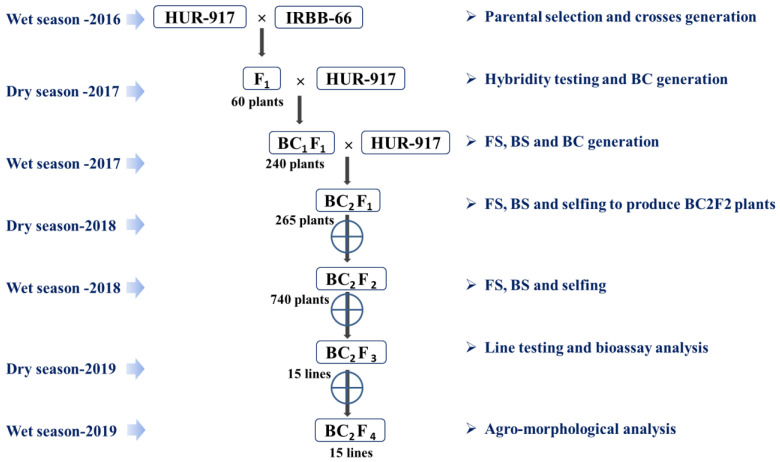
Marker-aided breeding strategy followed for the development of improved HUR 917.

**Figure 2 plants-12-01363-f002:**
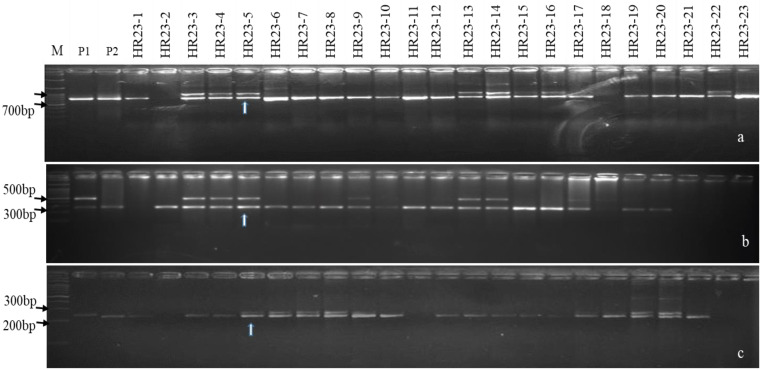
PCR amplification of BB resistance gene (s) in BC_1_F_1._ (**a**) Amplicons of the *Xa21* gene using pTA248 primer. (**b**) Amplicons of the *xal3* gene using xal3prom primer. (**c**) Amplicons of the *xa5* gene using RM122 primer; M, marker; P1, donor parent (IRBB66); P2, recurrent parent (HUR 917), Lanes 4–26 represent BC_1_F_1_ plants; vertical arrows indicate a positive plant heterozygote for all targeted genes.

**Figure 3 plants-12-01363-f003:**
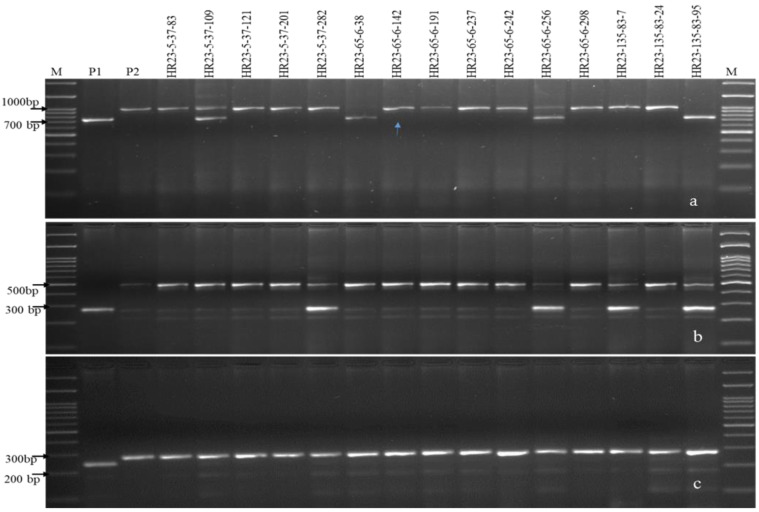
PCR amplification of bacterial blight resistant gene(s) in BC_2_F_2._ (**a**) Amplicons of the *Xa21* gene using pTA248 primer. (**b**) Amplicons of the *xal3* gene using xal3 prom primer. (**c**) Amplicons of the *xa5* gene using RM122 primer; M, marker; P1, RP (HUR 917); P2, donor parent (IRBB66), Lanes 4–18 represent NILs, vertical arrows indicate positive plant homozygous/heterozygous for targeted genes.

**Figure 4 plants-12-01363-f004:**
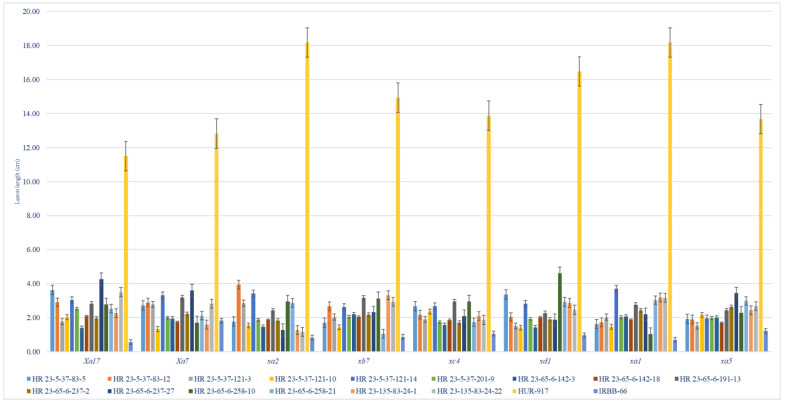
Extent of BB resistance in parents and derivative NILs (BC_2_F_3_) inoculated with 8 virulent *Xoo* isolates after 21 days of inoculation.

**Figure 5 plants-12-01363-f005:**
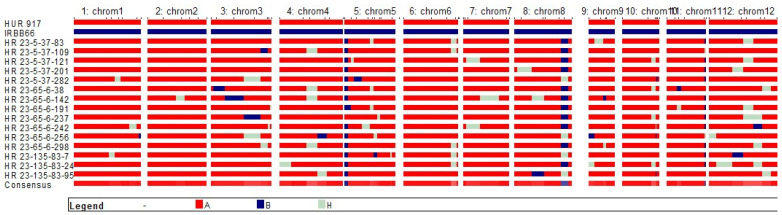
Graphical representation of genomic contributions from parents in NILs derived from the cross between HUR-917 (RP) and IRBB66 (Donor), NILs: 1–15, Legend A-represent RP, B-donor and H-heterozygosity; Chromosome 5, 8 and 11 harbors R genes, *xa5*, *xa13* and *Xa21*, respectively.

**Figure 6 plants-12-01363-f006:**
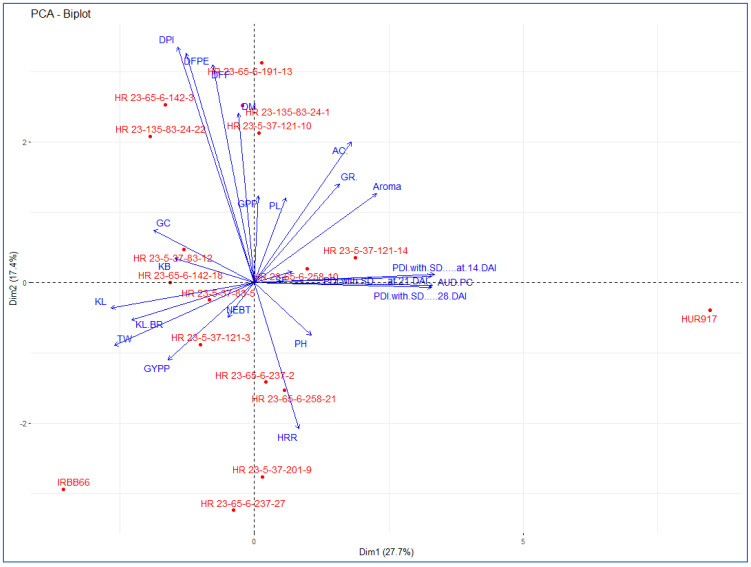
Biplot diagram of 21 traits including yield, quality and other agro-morphological traits in the parents and NILs.

**Table 1 plants-12-01363-t001:** Details of parental lines, their pedigrees, maturity duration and zone of adoption.

Genotype	Pedigree	Special Features	Yield (q/ha)	Duration (Days)	Recommendation for Cultivation
HUR 917	Dehradun Basmati Selection-13	High yielding, semi dwarf plant stature (105–110), resistance to lodging and neck blast disease and good grain quality. Susceptible to BB.	42–45	135–140	Released for cultivation in UP during 2015
IRBB66	NIL-IR 24	NIL of IR24, harbours 5 BB resistant genes (*Xa21 + xa13 + Xa7 + xa5* + *Xa4*) and broad spectrum of BB resistant against all prevalent philli-races.	55–60	135–140	NIL of IR 24, developed at IRRI, Philippines

**Table 2 plants-12-01363-t002:** Gene specific/linked SSR markers used for introgression of the corresponding gene in HUR 917 background.

Gene/QTL	Chr. No.	Marker/ Primer	Physical Position (Mb)	Type	Marker/Primer/Sequence	Reference
*xa5*	5	RM122	1.65	FS	Forward-5′gcactgcaaccatcaatgaatc3′ Reverse-5′cctaggagaaactagccgtcca3′	[35,36]
		RM17941	3.40	RS	Forward-5′gcctcgaagaaccagtagaacagc3′ Reverse-5′cttgtcttctcctcctcctgtgc3′	
*xa13*	8	xa13prom	26.81	FS	Forward-5′ggccatggctcagtgtttat3′ Reverse-5′gagctccagctctccaaatg3′	[3,20,36]
		RM23356	24.20	RS	Forward-5′gcctccaacagatctcctatctgg3′ Reverse-5′tttggcgctaatgagagattgg3′	
		RM22914	30.90	RS	Forward-5′ccaatcattaacccctgagc3′ Reverse-5′gccttcatgcttcagaagac3′	
*Xa21*	11	pTA248	22.60	FS	Forward-5′agacgcggaagggtggttcccgga3′ Reverse-5′agacgcggtaatcgaaagatgaaa3′	[7,36,37]
		RM26969	21.50	RS	Forward-5′ctcacacttgcaacatcctagc3′ Reverse-5′aaggctctagttggtgaagacc3′	
*BADH2*	8	FMbadh2-E7	82.80	FS	Forward-5′ggttgcatttactgggagtt3′ Reverse-5′cagtgaaacaggctgtcaag3′	[38]

**Table 3 plants-12-01363-t003:** Brief summary of crosses and RP genome recuperation in the NILs of HUR 917.

Generation	Total Plants Raised	Gene Positive Plants ¶	Plants Advanced	% RPG Recovery	Selection Criteria
F_1s_	60	52	05	*	Hybridity testing with gene linked markers
BC_1_F_1_	240	18	06	69.51–81.71	FS, BS, Phenomics and bioassay
BC_2_F_1_	265	33	03	84.76–91.46	FS, BS, Phenomics and bioassay
BC_2_F_2_	740	15	03	92.07–96.95	FS, BS, Phenomics and bioassay
BC_2_F_3_	15 families (each 150 plants)	15 families (15 plants)	15 NILs	>94.0	Phenomics and bioassay analysis
BC_2_F_4_	15 NILs	15 NILs	15 NILs	>94.0	Phenomics and bioassay analysis

* Not estimated; RPG, recurrent parent genome; ¶ number of plants/lines positive for *Xa21* + *xa13* + *xa5*; FS, foreground selection; BS, background selection.

**Table 4 plants-12-01363-t004:** Disease reaction (LL after 21 days of inoculation) of NILs (BC_2_F_3_) carrying introgressed resistance genes in different combinations against 8 different races of *Xoo*.

Breeding Lines/*Xoo* Isolate/Score	Gene Combination	LL in cm (Mean ± Standard Error)									
		*Xa17*	*Xa7*	*xa2*	*Xb7*	*Xc4*	*xd1*	*xa1*	*xa5*	MLL	Disease Reaction
HR 23-5-37-83-5	*Xa21 + xa13 + xa5*	1.63 ± 0.23	2.72 ± 0.17	1.78 ± 0.13	1.70 ± 0.22	2.67 ± 0.12	1.37 ± 0.90	1.63 ± 0.15	1.93 ± 1.36	2.43	R
HR 23-5-37-83-12	*Xa21 + xa13 + xa5*	2.90 ± 0.36	2.88 ± 0.46	3.94 ± 0.14	2.67 ± 0.25	2.17 ± 0.55	2.04 ± 0.31	1.72 ± 0.20	1.90 ± 0.20	2.53	R
HR 23-5-37-121-3	*Xa21 + xa13 + xa5*	1.77 ± 0.15	2.78 ± 0.13	2.85 ± 0.18	2.03 ± 0.49	1.90 ± 0.62	1.51 ± 0.87	2.03 ± 0.51	1.53 ± 0.87	2.05	R
HR 23-5-37-121-10	*Xa21 + xa13 + xa5*	2.03 ± 0.47	1.34 ± 0.37	1.55 ± 0.18	1.43 ± 0.21	2.37 ± 0.15	1.42 ± 0.16	1.47 ± 0.25	2.17 ± 0.32	1.72	R
HR 23-5-37-121-14	*Xa21 + xa13 + xa5*	4.03 ± 0.48	3.32 ± 1.51	3.42 ± 0.60	2.63 ± 0.71	2.67 ± 0.81	2.83 ± 0.65	3.70 ± 1.61	1.97 ± 1.16	2.95	R
HR 23-5-37-201-9	*Xa21 + xa13 + xa5*	2.53 ± 0.31	2.00 ± 1.19	1.87 ± 0.83	2.06 ± 1.35	1.74 ± 0.51	1.93 ± 0.35	2.03 ± 1.27	1.99 ± 1.57	2.02	R
HR 23-65-6-142-3	*Xa21 + xa13 + xa5*	1.40 ± 0.69	1.95 ± 1.59	1.47 ± 1.69	2.20 ± 0.26	1.57 ± 0.65	1.43 ± 1.53	2.07 ± 0.42	2.00 ± 0.53	1.76	R
HR 23-65-6-142-18	*Xa21 + xa13 + xa5*	2.09 ± 0.36	1.75 ± 0.15	1.87 ± 0.21	2.06 ± 0.12	1.86 ± 0.22	2.03 ± 0.25	1.87 ± 0.21	1.70 ± 0.10	1.90	R
HR 23-65-6-191-13	*Xa21 + xa13 + xa5*	2.83 ± 0.55	3.19 ± 0.54	2.43 ± 1.07	3.17 ± 1.07	2.96 ± 1.34	2.27 ± 0.21	2.77 ± 0.15	2.43 ± 1.01	2.76	R
HR 23-65-6-237-2	*Xa21 + xa13 + xa5*	1.97 ± 0.38	2.22 ± 0.59	1.83 ± 0.12	2.17 ± 0.35	1.70 ± 0.66	1.90 ± 0.20	2.43 ± 1.79	2.63 ± 0.15	2.11	R
HR 23-65-6-237-27	*Xa21 + xa13 + xa5*	4.27 ± 1.30	3.60 ± 0.53	1.27 ± 0.21	2.32 ± 0.53	2.10 ± 0.30	1.87 ± 1.42	2.20 ± 1.23	3.44 ± 0.92	2.63	R
HR 23-65-6-258-10	*Xa21 + xa13 + xa5*	2.78 ± 0.19	1.71 ± 0.11	2.95 ± 0.22	3.13 ± 0.49	2.95 ± 0.62	4.61 ± 0.87	1.03 ± 0.53	2.28 ± 0.82	2.68	R
HR 23-65-6-258-21	*Xa21 + xa13 + xa5*	2.53 ± 0.31	2.12 ± 1.19	2.87 ± 0.83	1.06 ± 1.35	1.74 ± 0.51	2.93 ± 0.35	3.03 ± 1.27	2.99 ± 1.57	2.41	R
HR 23-135-83-24-1	*Xa21 + xa13 + xa5*	2.27 ± 1.3	1.6 ± 0.53	1.27 ± 0.21	3.32 ± 0.53	2.10 ± 0.30	2.87 ± 1.42	3.20 ± 1.23	2.44 ± 0.92	2.38	R
HR 23-135-83-24-22	*Xa21 + xa13 + xa5*	3.5 ± 1.32	2.82 ± 1.41	1.17 ± 1.23	2.93 ± 1.12	1.87 ± 1.70	2.47 ± 1.45	3.17 ± 0.35	2.67 ± 0.50	2.58	R
HUR 917	*xa21 + Xa13 + Xa5*	11.50 ± 1.32	12.82 ± 1.41	18.17 ± 1.23	14.93 ± 1.12	13.87 ± 1.70	16.47 ± 1.45	18.17 ± 0.35	13.67 ± 0.50	14.95	S
IRBB66	*Xa21 + xa13 + Xa7+ xa5 + Xa4*	0.57 ± 0.21	1.83 ± 0.35	0.83 ± 0.06	0.87 ± 0.31	1.07 ± 0.55	0.97 ± 0.21	0.70 ± 0.10	1.23 ± 0.15	1.01	R

R, resistant; MR, moderately resistant; S, susceptible; MLL, mean lesion length (cm).

**Table 5 plants-12-01363-t005:** ANOVA of parents and BC_2_F_3_ and BC_2_F_4_ (pooled) derivatives of HUR 917/IRBB66 for morpho-agronomical traits.

Source	Replication	Genotype	Residual
DF	2	16	32
DPI	1.82	23.051 ***	0.82
DFPE	0.94	18.936 ***	0.90
DFF	1.91	11.676 ***	0.86
DM	2.43	42.824 ***	1.58
PH	0.09	23.435 ***	0.21
NEBT	0.11	3.665 ***	0.20
PL	0.28	6.033 ***	0.12
GPP	0.23	176.052 ***	5.38
SF	0.49	53.479 ***	0.37
GYPP	0.17	7.129 ***	0.16
TW	0.13	16.726 ***	0.17
KL	0.02	0.707 ***	0.01
KB	0.82	0.010 ***	0.02
KL BR‘	0.21	0.193 ***	0.01
HRR	0.21	155.557 ***	0.21
AC%‘	0.17	6.481 ***	0.07
GC	0.25	282.764 ***	0.20
Aroma	0.02	0.813 ***	0.02
PDI with SD (%) at 14 DAI‘	0.02	132.146 ***	0.01
PDI with SD (%) at 21 DAI‘	0.32	358.110 ***	0.02
PDI with SD (%) 28 DAI‘	0.01	516.47 ***	0.01
AUD PC‘	0.42	112.199 ***	1.00
GR%‘	0.03	155.516 ***	0.04

*** significant at *p* ≤ 0.001 probability level. Note: DFF, days to 50% flowering; DM, days to maturity; PH, plant height; NETPP, number of effective tillers per plant; PL, panicle length; NGP, number of grains per panicle; TW, test weight; GYPP, grain yield per plant; KL, kernel length; KB-kernel breadth; KLBR, kernel length/breadth ratio; HRR, head rice recovery; AC, amylose content; GC, gelatinization content; PDI, percent disease index; DAI, days after inoculation; AUDPC, area under disease progress curve; GR, genome recovery.

**Table 6 plants-12-01363-t006:** Duncan’s Multiple Range test (DMRT) for analysis of pairwise differences among parents and NILs of HUR 917/IRBB66.

Genotypes	DFF	DM	PH	NEBT	PL	GYPP	TW	SF	HRR	KL	KB	KLBR	GC	AC %	Aroma	PDI at 21DAI	AUDPC
HR 23-135-83-24-1	110 b	136 b	103 h	11 bc	25.1 a	16.8 g	17.3 c	82.51 g	64 h	5.9 abcd	1.8 bc	3.3 cde	44 e	24.03 a	2 b	7.19 e	103.35 e
HR 23-135-83-24-22	110 b	136 b	109.6 b	11 b	23.7 cde	19.6 cd	20.27 a	88.36 bc	61 j	6.5 a	1.8 c	3.6 ab	58.97 a	22.77 c	2 b	7.26 d	97.49 g
HR 23-5-37-121-10	110 b	136 b	107.4 c	9 fghi	25.5 a	21.2 a	14.35 f	85.36 e	66 g	5.4 bcd	1.7 def	3.1 def	36 h	22.83 c	2 b	4.69 k	64.1 j
HR 23-5-37-121-14	108 cd	128 f	101.6 i	10 defg	23 ef	16.8 g	13.57 g	88.11 bc	77 b	5.1 d	1.7 ef	3 fg	38.11 g	24.04 a	2 b	6.99 g	122.29 c
HR 23-5-37-121-3	106 e	134 bcd	102.9 h	9 hi	24.8 ab	20.7 ab	20.13 a	86.63 d	63 i	6.3 ab	1.7 de	3.6 ab	28.37 m	21.75 d	2 b	4.06 m	59.59 k
HR 23-5-37-201-9	107 de	126 g	106.8 cd	13 a	23.4 def	18 f	18.38 b	88.37 bc	67 f	5.9 abcd	1.7 f	3.5 b	33.03 k	20.38 e	2 b	4.88 i	98.36 g
HR 23-5-37-83-12	110 b	131 e	104.8 ef	9 ghi	22.9 efg	18.9 de	18.77 b	82.74 g	59 k	6.4 a	1.7 def	3.7 a	38.98 f	20.14 ef	2 b	7.89 b	105.31 d
HR 23-5-37-83-5	108 cd	136 b	107.3 c	10 def	20.9 h	19.2 d	16.73 cd	84.09 f	69 de	6.2 abc	1.9 a	3.4 c	34.97 i	22.76 c	2 b	3.38 p	77.06 h
HR 23-65-6-142-18	108 cd	132 de	106.9 cd	10 cd	25.5 a	16.8 g	20.13 a	90.32 a	60 j	6.4 a	1.8 c	3.6 b	51.53 c	20.45 e	2 b	3.88 n	53.52 l
HR 23-65-6-142-3	112 a	136 b	105.3 e	11 bc	24.3 bc	18.2 ef	20.23 a	89.15 b	64 h	6 abc	1.9 a	3.3 cde	34.01 j	22.1 d	2 b	4.2 l	36 m
HR 23-65-6-191-13	110 b	141 a	103.8 g	10 cd	22.9 efg	16.7 g	16.43 d	88.21 bc	54 l	5.4 bcd	1.7 d	3.1 ef	55.32 b	23.46 b	2 b	7.47 c	154.34 b
HR 23-65-6-237-2	106 e	134 bc	105.4 e	9 efgh	24 cd	20.1 bc	18.31 b	87.69 cd	79 a	5.7 abcd	1.7 de	3.3 cd	31.04 l	23.44 b	2 b	3.3 q	65.08 j
HR 23-65-6-237-27	104 f	127 fg	106.4 d	11 bc	23 ef	19.4 cd	20.16 a	90.68 a	69 d	6.1 abc	1.8 abc	3.3 c	45.03 d	21.85 d	2 b	5.17 h	77.91 h
HR 23-65-6-258-10	109 bc	134 bcd	104.8 ef	9 i	22.1 g	18.3 ef	16.48 d	76.6 h	73 c	5.1 d	1.8 bc	2.8 h	36.13 h	22.76 c	2 b	7.16 f	100.38 f
HR 23-65-6-258-21	107 de	132 cde	104.1 fg	9 fghi	21 h	18 f	15.66 e	88.55 bc	79 a	5.4 cd	1.7 f	3.2 de	37.9 g	20.13 ef	2 b	4.77 j	64.48 j
HUR 917 (RP)	107 cde	135 b	112.8 a	10 cde	25 ab	7.1 h	14.27 f	87.69 cd	68 e	5.1 d	1.7 de	2.9 gh	28.03 m	23.69 ab	3 a	49.91 a	873.19 a
IRBB66 (donor)	108 bcd	136 b	109.3 b	11 bc	22.7 fg	20.8 a	20.53 a	76.71 h	73 c	6.2 abc	1.8 ab	3.4 c	55.27 b	19.77 f	0 c	3.41 o	73.74 i

Values having common letter(s) in a column do not differ significantly at *p* ≤ 0.05 as per DMRT.

## Data Availability

Not applicable.

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
