# Peer review of "Marker-Assisted Improvement for Durable Bacterial Blight Resistance in Aromatic Rice Cultivar HUR 917 Popular in Eastern Parts of India"

_plants, 2023, doi:10.3390/plants12061363_

Round 1
Reviewer 1 Report
Comments to the author(s)
The present study demonstrated the introgression of three BB-resistant genes in commercial aromatic rice through a marker-assisted improvement method. The research design was good, the analyses were well performed, and the results were also well prepared in a comprehensive description. The findings appeared to be useful and supportive in further deep study of specific genomic sites/regions related to bacterial blight disease. Still, the following minor concerns listed below are suggested to be improved.
Minor Concerns
Lines 42-47 Add citation(s)
Lines 50-52 Add citation(s)
Line 126-129 The sentence is describing BC1F1 plants. But the supporting information is too much (some are wrong, e.g., Table 4 is for BC2F2) and some are very general (not specific). Writing the descriptions with supporting information is good, but it should be better to point out the specified data with its corresponding supported figures or tables.
Line 172 It is also the same writing style with unrelated supporting information. Supl. Fig. 2 is supported and check whether it is confirmed or not.
Line 141 and 148 For gel images, it was noticed only 3 main figures including one supplementary. Surprisingly, the donor parent position (well/lane position) is always changing figure by figure. It is curious to know that it changed for some purposes. Although it’s over, the format should be in uniformity (because only two parents in the whole study).
Line 141 In figure 2(a), the product sizes for P1 and P2 look like the same. Please check and make it confirm for reasonable display the bands are clear though.
Lines 238 and 239 In figure 6, the resulting information is missing to be explained and described more. The figure is just added in line 205, like supporting information. The authors should detail more specifying some important traits and variables together with their vector values.
Abbreviations
Almost all the tables and figures are displayed with abbreviations. Every single Abbr. should be provided as notes or in captions. OR all Abbrs. for the whole manuscript should be provided on one separate page. Please check others along with the manuscript and select the best way of description.
Author Response
Response to the reviewer’ comments
Authors thank the Editor for arranging qualified Reviewers and providing us with the opportunity to re-revise our manuscript. As suggested, we have performed corrections in the revised manuscript. The corrected portions (reviewer1) are highlighted with yellow colour. We believe that the quality of the manuscript has now improved substantially. And, we hope that the revised version is acceptable for publication in the journal “Plants’.
Reviewer-1:
Comment -1: Lines 42-47 Add citation(s)
Response 1: Citation is added
Comment -1.2. Lines 50-52 Add citation(s)
Response 1.2. Citation is added
Comment 3: Line 126-129 The sentence is describing BC1F1 plants. But the supporting information is too much (some are wrong, e.g., Table 4 is for BC2F2) and some are very general (not specific). Writing the descriptions with supporting information is good, but it should be better to point out the specified data with its corresponding supported figures or tables
Response 3. As suggested, sentence is corrected with proper support.
Comment 4: Line 172 It is also the same writing style with unrelated supporting information. Supl. Fig. 2 is supported and check whether it is confirmed or not.
Response 4. Corrected as suggested.
Comment 5: Line 141 and 148 For gel images, it was noticed only 3 main figures including one supplementary. Surprisingly, the donor parent position (well/lane position) is always changing figure by figure. It is curious to know that it changed for some purposes. Although it’s over, the format should be in uniformity (because only two parents in the whole study).
Response 5. No it is not purpose based, donor’s lane should come first, it is just mistakenly done by student, now correction is not possible. Sure, we will remember it in future works.
Comment 6. Line 141 In figure 2(a), the product sizes for P1 and P2 look like the same. Please check and make it confirm for reasonable display the bands are clear though.
Response 6. It happened mistakenly, RP DNA is placed in P1 and P2 lanes, but its not big issue as in advanced generation figure (BC2F2) it is clear.
Comment 7. Lines 238 and 239 In figure 6, the resulting information is missing to be explained and described more. The figure is just added in line 205, like supporting information. The authors should detail more specifying some important traits and variables together with their vector values.
Response 7. PCA biplot is just supportive information to see the RP traits recovery, now it is explained in line no. 209-2011.
Comment – 8. Almost all the tables and figures are displayed with abbreviations. Every single Abbr. should be provided as notes or in captions. OR all Abbrs. for the whole manuscript should be provided on one separate page. Please check others along with the manuscript and select the best way of description.
Response 8. As per journal’s guideline, abbreviations are displayed, details are mentioned when it occurs first.
We hope that the revised version is acceptable for publication in the journal ‘Plants’.

Reviewer 2 Report
A well-done research project regarding a "genetic shortcut" (my words) toward enhanced, sustainable rice resistance to bacterial blight. Well written, straightforward to follow. I have made several comments in the text; most indicate that I positively received your work. One minor concern--related to modest yield depression for your RP, in spite of heavy disease infestation. Perhaps you can comment on that modest issue?

Author Response
Response to the reviewer-2’ comments
Authors thank the Editor for arranging qualified Reviewers and providing us with the opportunity to revise our manuscript. As suggested, we have incorporated all suggestions in the revised manuscript. The corrected portions are highlighted with sky blue colour. We believe that the quality of the manuscript has now improved substantially. And, we hope that the revised version is acceptable for publication in the journal “Plants’.
Comment 1. One minor concern--related to modest yield depression for your RP, in spite of heavy disease infestation. Perhaps you can comment on that modest issue?
Response. It is due to typing error, in data table 16.8g GYPP is typed instead of 6.8g which created mis-interpretation of DMRT value i.e. 17.0g instead of 7.1h. Now it is corrected, reanalysed and interpreted.
